# Survey and Measurements of Indoor Environmental Quality in Urban/Rural Schools Located in Romania

**DOI:** 10.3390/ijerph191610219

**Published:** 2022-08-17

**Authors:** Tiberiu Catalina, Stefan Alexandru Ghita, Lelia Letiția Popescu, Răzvan Popescu

**Affiliations:** 1Faculty of Building Services, Technical University of Civil Engineering, 021414 Bucharest, Romania; 2National Institute for Research-Development in Construction, Urbanism and Sustainable Territorial Development—INCD URBAN-INCERC, 400524 Cluj-Napoca, Romania; 3Faculty of Architecture and the Built Environment, Delft University of Technology, 2628 CD Delft, The Netherlands

**Keywords:** indoor environmental quality, school buildings, survey

## Abstract

Taken as a whole, the indoor environmental quality (IEQ) is a concept that deals not only with thermal conditions, but it also goes much further, because it includes indoor air quality (IAQ), illuminance or acoustic comfort. Among the different categories of buildings, schools are one of the most important in society especially because future generations are spending more than 6 h/day and ensuring them a healthy and comfortable environment must be top priority. The purposes of this research were to better understand school children’s IEQ preferences and needs in classrooms and to compare these among rural versus urban respondents. To reach this goal, a simple yet complete survey was proposed and, along with measurements, multiple conclusions were made. The methodology proposed was to reach a large sample of respondents to give more credibility and precision to the analysis. The results are based on the responses of 790 occupants both from urban and rural environments. Lack of ventilation, cooling, low or too high temperatures during winter/summer periods and a misappropriate sizing or piloting of the heating system are some of the issues found during the survey. The paper tackles several issues at once, helping to paint a more holistic image of the problems encountered in school classrooms. Optimal thermal comfort is not met during the cold season in any of the schools under investigation. The old rural schools were found to be the worst performing buildings compared to urban schools where due to recent investments in refurbishment the winter thermal comfort was enhanced. During the study, it was observed that one of the major IAQ problems consisted in elevated levels of CO_2_ or particulate matter especially for the schools situated in the city. Noise was reported as an issue only for the schools situated next to national roads while visual comfort was found to be acceptable for more than 94% of respondents from all regions.

## 1. Introduction

The indoor environment has an important influence on the comfort, health, attention spans and learning results of its young occupants, especially in schools, buildings of significant importance to the community. Discussing the correlation between IEQ (thermal, acoustic, visual, IAQ) and the well-being of students is of high interest. One important part of the IEQ is the thermal comfort that can severely affect the occupants in terms of health, productivity or simply well-being [1]. In many cases, the use of surveys to assess the thermal comfort sensation is one simple but reliable method, especially if the number of respondents is large [2]. Along with surveys, researchers are also measuring multiple indoor parameters such as air temperature, humidity, air velocity, etc. [3,4]. One major advantage of the use of field studies is that it is a realistic approach to the evaluation of thermal comfort in comparison to laboratory conditions. In schools, the occupants follow a certain dynamic behavior pattern, meaning that, in most cases, for almost one hour they are sitting while for ten minutes are playing or doing diverse activities. Thus, their metabolic rate follows multiple stages—from high concentration to the enthusiasm during the breaks. Moreover, their normal learning activities including writing, listening, speaking, modifying clothing level, window opening or use of shading blinds can severely affect the thermal sensation [5]. As children have a lower weight mass, less surface exposed to the environment or fluctuating metabolism, the main thermal comfort criteria are difficult to use for them [6]. In multiple research studies on schools, the applicability of thermal standards was evaluated [7,8,9]. The predicted mean vote (PMV) was calculated for 79 children by Ter Mors et al. [10] in non-air-conditioned classrooms. This type of evaluations dates back to 1970s [11,12]. The assessment of the thermal environment is usually carried out by means of PMV and PPD indices. This approach was used in 200 Italian classrooms with over 4000 students by means of a special survey [13]. 

A similar method was employed in an old Spanish school where a comparison between PMV, TSV and PPD was made for both the educators and the pupils [14]. Another interesting study was realized in the Slovak Republic where the research took place in five classrooms in a special school. Using surveys of 34 pupils and measurements, the research team assessed the PMV/PPD but also CO_2_ level, humidity and noise. Their results confirmed the close relationship between IEQ and occupants’ wellbeing [15]. The classroom characteristics could also influence the health and comfort of school children as was pointed out by [16] with a survey of 1311 occupants and the Classroom Symptom Index and Classroom Comfort Index were proposed. 

In addition to maintaining an appropriate thermal comfort and IAQ, classrooms should also secure the best learning environment for children in terms of the acoustic performance. Guidelines have also been published by several organizations, with recommendations on the acoustic environment of both the outdoor and indoor spaces of schools. High background noise levels in occupied classrooms have been long identified as factors negatively affecting the academic performance of students. 

The efficiency of communication and, hence, the efficiency of the learning environment, is measured by the acoustic conditions of the classrooms. In multiple studies, it is mentioned that the speech to noise ratio (SNR) should be at least +15 dB(A) at the child’s ears to achieve appropriate acoustical conditions in an educational setting. Unfortunately, these guidelines are rarely followed. Background noise levels in classrooms measured around the world usually exceed 35 dB(A) [17]. High levels of noise in classrooms make students prematurely tired, lowering their cognitive abilities with regard to paying attention and understanding the content of their classes [18]. Around the world, several studies have focused on the topic of acoustic comfort including assessment of speech intelligibility, impact of materials on the reverberation time and noise propagation in either schools or colleges [19,20]. 

One study conducted in the Netherlands showed that 85% of 335 school children have reported noise problems [21]. Another direction that studies undertook was related to the sensitivity to noise of pupils, which was investigated by Zannin and Marcon [22]. 

One of the main issues associated with visual comfort is daylight. It is a key component that supports human activity and health. Thus, significant scientific studies have been performed on the effects of daylighting in classrooms, primarily focusing on the potential correlation between daylighting and student productivity [23]. Although much emphasis is placed by some authors and governmental institutions on the importance of daylighting, the need for integrated systems of day- and artificial lighting is broadly accepted [24]. Overall, the description of visual comfort is less standardized than the more famous thermal comfort. There are multiple reasons for this: it is clearly dependent on the outdoor illuminance levels, type of window frame/glazing, type of visual task or characteristics of the indoor space (e.g., reflectance coefficients) [25]. The Standard EN 1264-1 establishes the fundamental rules for artificial lighting in indoor workplaces, including schools, and tackles the glare issues, encouraging the integration between natural and artificial lighting [26]. Also the standard *EN 16798-1 is a good review of indoor parameters to be achieved in buildings* [27].

One important source of data on the IEQ and the impact on thermal comfort, indoor space contentment and self-reported performance comes in the form of the actual occupants of the building. Moreover, important feedback may be provided to architects, designers and building owners to assess building features and technologies by means of occupant satisfaction and perception of the environment responses. Poor indoor environmental quality in classrooms may be a risk for health symptoms and cause absence from school as was found by Turunen et al. [28] in 355 elementary schools in Finland. Post-occupancy evaluation (POE) [29] delineates a process aimed at assessing buildings once they have been occupied in order to improve the existing conditions and as a guide for the design of future buildings. A feedback loop can be created with POE that architects, planners and managers can use to learn how different building design features and technologies may affect occupant comfort, satisfaction and productivity. Some of the tools, which the POE process makes use of, include surveys, cohort studies, observations and task performance tests. All of these can be used either alone or in combination with quantitative physical measurements. The drawback of this procedure is that currently there is no standardized method to survey buildings occupants, and especially educational establishments [3]. 

In a study conducted in Budapest, a complex method was implemented to examine and explore the use of natural ventilation in classrooms using the windows. The approach followed two directions (survey and experimental measurements) with the aim to identify environmental, background or occupants’ need for better air quality [30]. 

The use of surveys was also used to assess the IEQ in an Irish university where the authors made clear that precaution must be used so as not to degrade human health in favor of energy savings [31]. 

Surveying is seen as the simplest and least expensive method for evaluating IEQ concerns in a building [32]. Occupant satisfaction is ultimately the primary interest of the building owner/operator regardless of physical IEQ conditions and thus many survey tools are available for studying perceived comfort. Schiavon and Peretti’s [3] review of IEQ surveys provides a historical account of IEQ surveys. The two most widely used survey methods are that of Building Use Studies Ltd. [33] and the CBE Berkley University one [34]. 

Nonetheless, the subjective nature of surveys and range of opinions that can be expressed for similar IEQ physical conditions complicate the use of surveys as the only tool for evaluating building IEQ performance. Furthermore, surveys do not always capture IEQ issues that may have energy implications (e.g., over-lighting or economizer operation) and have incomplete diagnostic capability. Nicol and Wilson [1] discuss other issues associated with surveys, including: difficulty finding a representative period for surveying [2] and interpreting the results; and Peretti and Schiavon [3] investigated the main questions that must be answered [35].

The first critique can be partially addressed by carrying out “right-now” evaluations at various moments of the day/week/month/year, though this can potentially lead to “survey fatigue” [36]. “Right-now” surveys ask about conditions when the survey is given, as opposed to long-term surveys that ask occupants to summarize their overall satisfaction for the past week, month or year. The second critique refers to the lack of clear guidelines for practitioners on how to transform subjective measures into standardized limits of environmental parameters. For example, how should visual comfort satisfaction scores be interpreted in terms of light levels and glare ratios? The third critique refers to the complicated nature of survey questions, which can greatly affect the answers received and lead to biased, or otherwise inaccurate, results which complicate comparisons between surveys. Other factors, including psychological and physiological states, and cultural and economic differences, are not typically accounted for in surveys [37]. Benchmarking requires the static nature of the two most widely used occupant survey databases (CBE and BUS), making it difficult to edit existing questions or implement new questions that decrease bias and improve accuracy. Catalina and Iordache [38] analyzed the IEQ index using simulations and obtained fast regression models to predict different indoor parameters. 

This paper follows the same logical lines as the previous studies with a focus on the comparison of IEQ between rural and urban schools. Furthermore, the study deals with a specific type of school, specifically the ones built in the communist period in Romania, thus the results are intriguing and bring valuable information to the existing international literature.

## 2. Methodology

### 2.1. Survey Description

As was made clear in this paper’s Introduction, indoor environment quality (IEQ) plays a role of paramount importance in the educational building sector. Hence, regardless of whether the establishments under discussion are primary or secondary schools or even universities, one thing is certain: establishing an optimal indoor environment is crucial. Given that the present investigation is the very first of its kind in Romania, as it tackles multiple urban/rural school environments with a thoroughly scientific approach, its outcomes promise to provide interesting insights into the field of IEQ in schools. Furthermore, urban respondents outweigh their rural counterparts (around 80% of the survey respondents come from a city), and the following explanation can be found. First, this phenomenon is consistent with what is currently being observed at a national level; more and more rural communities are losing their inhabitants in search of a better life in the bigger cities (and especially the capital of Romania, Bucharest). 

From the outset, the goal of the research project promised to be a challenging one due to the ambitious target set: providing legislative recommendations on the improvement of indoor environment quality in schools, all whilst increasing energy efficiency and integrating renewable energy sources. For this, a systematic approach had to be considered. Ergo, before any corrective measures could be taken, it was important to first assess and understand the current situation pertaining to Romanian schools. In answering this question, there are two main directions (mentioned earlier) that the investigator can follow: a subjective (surveys) and an objective (experimental measurements) approach. For the current study, both methods were used concomitantly to effectively correlate questionnaire respondents’ perception with actual recorded data. 

The authors of this paper have selected multiple schools based on their availability for survey, as in many cases it is challenging to find willing respondents, but also schools from rural and urban environments for comparison purposes. Eight schools have been selected, of which five are from urban environments and the rest are rural. The interest in participating in this research was higher for the urban respondents but nevertheless the three rural schools cooperated well with the research team. 

Accordingly, the subjective approach consisted of over 790 pupils completing a survey during the regular lesson period while the physical measurements were being carried out. This enabled assessing the respondents’ level of satisfaction by means of a certain scale associated with the multiple-choice questions. The wording of the questions was designed to be easily comprehensible for all age categories starting from 11 years up to 22 years. Thermal comfort was studied through inquiries on the perceived temperature satisfaction (both present and winter/summer). The pupils received several questions regarding the thermal comfort during winter/summer seasons and their responses are based on their previous experiences during previous years spent in the same school/classroom, while the question regarding the temperature satisfaction is based on the response while filling out the survey. 

Indoor air quality questions focused on the sensed level of dust, the cleanliness of the air and a general need of the occupants to open the windows or not. Visual comfort referred to seeing whether enough light was reaching the working space of the pupils. Lastly, the fourth main category was related to acoustic comfort. Respondents were asked how they dealt with noise coming from the exterior and whether they could hear the professor properly (to check issues about reverberation problems). Three additional subquestions were asked, giving insight into the age, gender and clothing of the respondents. A detailed account of the survey used can be found in the Annex of this paper. 

Although subjective in nature, this analysis method is relevant because it focuses on the users that decide whether an optimal IEQ is achieved or not. From this, the intermediate goals of the research were reached. Along with understanding whether thermal comfort is achieved throughout the year, perceived air quality and acoustic and visual comfort are also covered in the current survey. Additionally, grasping what children actually wear during school hours and how they have adapted to certain conditions (sometimes inadequate illuminance levels and low ambient air temperatures) is of vital importance if improved construction guidelines are to be elaborated. An additional objective in this paper is whether proper classroom ventilation is ensured using CO_2_ and IAQ indications as well as solid particles’ concentration recordings. The data were collected in all schools and a comparison with the national directives took place. The survey was carried out during the autumn season, more exactly at the end of September and lasted approximately for 2 weeks. The respondents answered questions related to the period of the survey but also more general questions about the main important indoor parameters for their comfort. In meeting these goals/objectives, the survey campaign was supported by experimental spot measurements carried out at the same time the surveys were being filled in. Moreover, in a subsequent section, the results of these measurements as well as long-term recordings will be presented in greater detail. For now, the main criteria analyzed as well as the equipment employed will be described. To obtain a wide-ranging idea of what are main features of the indoor space, microclimatic data such as ambient air, mean radiant temperature, mean air velocity, relative humidity (interior and exterior), CO_2_ concentration, particulate level, illuminance and noise level were all recorded and processed. The instantaneous monitoring had a time step of either 1 min (TESTO 480 instrument) or 10 min for the other instruments used. The TESTO apparatus complies with the ISO 7726 [39] and ISO 7730 [40] standards and it collected data of humidity, air temperature, air velocity, CO_2_ level, radiant temperature and illuminance levels. The position of the sensors was set up in order not to perturb the educational activity or to distract the attention of the students. During the measurements, the placement of sensors next to doors, windows, artificial heat sources (e.g., heating radiator, video projector) or direct irradiation from the Sun was avoided as recommended by ISO 7726 [39]. The research team installed the equipment in almost all cases at least 30–45 min before the arrival of the occupants. The outdoor climatic conditions (air, humidity and CO_2_ levels) were measured with the CO_2_ m equipment. Details of the equipment including their range, resolution and accuracy are provided in Table 1. 

### 2.2. Schools Investigated

The survey campaign was conducted in eight schools ranging from primary to university level and spanning across both urban and rural environments. Table 2 presents the analyzed schools. The intention was to gain sufficient data so as to observe and understand the differences between the two environments which would then result in better adapted solutions. Five out of the eight educational buildings chosen for this research are located in Bucharest, while the remaining three are situated in Valcea County, in a semi-mountainous region (near Transylvania). A total of 790 survey responses were gathered out of which 478 (60.5%) came from male respondents and the remaining 312 (39.5%) from female respondents. For accounting purposes, the countryside made up 14.4% of all survey responses (8.2% male and 6.2% female out of the total survey population) while the remaining 85.6% came from the urban population (33.5% female and 52.3% male). What can be observed from this initial categorization is that in both urban (40% female and 60% male) and rural (42% female and 58% male) cases the gender distribution is similar with slightly more female students present in the countryside. This observation could also be influenced by the types of schools (and classrooms) the surveys were distributed in. For example, it is well known that the students of UTCB University are predominantly male, whereas, had the surveys been performed in a more humanities-oriented university, the results would have been the opposite way around. A clearer description of the schools surveyed can be found in the following table where Schools 1, 3, 4, 5, 7 and 8 are old and not thermally refurbished, School 6 is new (less than 3 years since its construction) and the only renovated school is School 2.

The above table brings forth a detailed account of which schools were surveyed as part of this initial stage of the research study. For ease of representation, each of the schools is given a code name (ranging from S1 to S8) which in later graphs will be further simplified. Overall, the initial observation yields that most schools have a balanced percent distribution of the total survey body. The only exception is School 4 (“*Anghel Saligny*” industrial high school) with over 50% of the total survey responses. The explanation for this phenomenon comes from the great collaboration had with this particular educational establishment. While at the other schools the surveys were personally distributed, the professors at School 4 agreed to be actively involved and handed the surveys to all of their pupils which resulted in a higher number of responses received. However, this should not be looked upon as perturbing factor, one that corrupts the conclusions of the survey, since this particular secondary school is representative of a large part of the educational building stock in Romania (both in terms of building shape, size, materials, heating system, heating control system, pupils attending, etc.).

Usually, a typical urban school in Romania has 18 to 24 classrooms, 2 laboratories (biology and chemistry), 3 specific classrooms (IT, German/English lessons), 1 library, 1 sport gym, 1 professors’ room, 1 medical room and 1 counselling room. On the other hand, in Romanian rural regions the schools are much smaller with 4 to a maximum of 12 classrooms, 1 medical room, 1 professors’ room and 1 IT room. The sports activities take place outside in almost all Romanian educational facilities. Figure 1 illustrates the facades of a rural school and on the right side that of a new/renovated building. There are multiple differences in terms of heating system (stove vs. wood/gas boiler, double glazed wood pane windows vs. PVC double pane windows, interior finishings—see Figure 2 for luminaire type and number, size of the room). 

Table 3 highlights the age and gender distribution of the surveyed population. The most important comments that can be made based on Table 3 data are that the largest part of the urban respondents fall in the interval between the ages of 14 and 18; on the other hand, the rural population is concentrated between the ages of 10 to 14 years. These observations are consistent with the general trend occurring at a national level. In countryside regions of Romania, we found only primary schools with less equipment (e.g., IT, laboratories) and that were overall inadequate for higher education, children generally study there up to the age of 14 when some of them go on to enroll in high school in the nearby cities. 

From the age of 14 to 18, pupils are enrolled in high schools with only a fraction deciding to go for a university degree. Although the latter comment does not entirely reflect the reality of university enrolments, it still manages to show a particular decline between those opting to follow undergraduate education and those content with a high school diploma. The following Figure 3 provides a visual representation of the remarks made.

## 3. Discussion

### 3.1. Thermal Comfort Assessment

After better understanding the background of the survey respondents, the attention will now shift to assessing their perceived thermal comfort. For this particular section, the survey contained three main questions and two additional subquestions. The first three inquired on the thermal perception of the respondents at the time of the survey, then focused on the summer and finally on the winter season. Each of these was assessed on a scale ranging from very cold to very warm with a just right/neutral response in the middle. For ease of manipulation, they were further distilled into a −2 +2 scale. The second two questions inquired on the perception of air currents and on the level of clothing insulation. The latter inquiry was important to both understand what students wear during certain seasons as well as understand how this affects their declared thermal comfort. Children were given the possibility to either choose from a list of predefined clothing types (see Table 4) or mention what they were wearing. Table 4 indicates which types of clothing/values were used in the survey/calculations. As for the air currents question, the intention was to find out whether children can perceive small indoor air wind speeds and whether this would impact the thermal comfort in any way. In the end, it was found that the overwhelming majority (75%) indicated that no air currents were felt. This conclusion could arise either from the fact that children cannot accurately perceive small air drafts or from the double insulated windows offering good overall air tightness. Hence, these results are inconclusive to draw a substantiated conclusion and were not used further.

#### 3.1.1. Influence of Clothing Insulation on the Perceived Thermal Comfort

With the data gathered from the responses on thermal sensation and clothing insulation, a number of graphs were obtained which offer some interesting insights. For increased clarity, clothing insulation is divided into four main intervals covering all clothing typologies used, from summer to extreme cold apparel. A number of statements can be made at this time: the first two intervals (0–0.5; 0.5–0.75 clo) are relatively balanced (40% compared to 37%), the third one (0.75–1 clo) is slightly smaller (20%), whilst the latter interval (1–1.5 clo) is barely represented (1.8%). Nonetheless, these findings are misleading in that they do not indicate the actual number of respondents for each of the thermal comfort replies. Thus, the following figure takes the information and provides further details. 

Figure 4 contains a considerable amount of information requiring further explaining. On the horizontal axis, the clothing insulation (expressed here in clo, where $1\;\mathrm{clo}=0.155\;\frac{W}{m^{2}K}$) is plotted, whilst on the vertical axis the thermal comfort perception is recorded (with values ranging from −2 to +2). As can be observed, the majority of the answers 586 ≡ 74% are at the 0 value, corresponding to a neutral (“just right”) sensation. This happens regardless of the clothing worn by the respondents at the time of the survey. Those that estimated that the environment was either cold (96 ≡ 12%) or warm (56 ≡ 8%) are a minority. Finally, the two extremes, very cold (8 ≡ 5%) and very warm (8 ≡ 1%) can be looked upon as either the outliers of the group (randomly filled in the surveys) or as extremely sensitive people. Overall, they do not influence the outcomes of the investigation. In general, it is accurate to say that at the time of the survey most students were satisfied with the classroom temperature. Furthermore, the way the respondents were dressed shows that on average 40% wore light summer clothes, 37% added an extra sweater, 20% added a jacket on top of the sweater and only 3% were excessively dressed. In conclusion, looked upon from a global perspective (twain urban and rural, male and female), it is safe to assert that classroom occupants are satisfied with the ambient temperature. A more detailed differentiation will be made later in the project when these initial results will be further refined. 

#### 3.1.2. Interpreting the Individual Survey Question Results

As previously indicated, the surveys were made up of three main questions. From the outset it decided to assess (based on subjective answers) thermal comfort both during the filling period of the survey but also during hot/cold seasons. Ergo, the occupants were asked to state how the classroom feels during both hot/cold seasons. With this information, conclusions will be drawn in the ensuing paragraphs.

Question 1: “How do you find the classroom now?”

Using this initial question, it was intended to gather information with regard to thermal comfort perception of the survey respondents for the period 30 of September to 8 of October. Based on the answers given, a detailed interpretation followed. The analysis was made for the urban and rural setting, then gender differentiated to see whether there are any similarities or differences when it comes to the perceived thermal comfort. On top of this, depending on the recorded instantaneous temperature, another classification resulted in a complete explanation of the procedure presented in the next paragraph.

The respondents were placed in a category depending on their age, with three intervals decided for the urban respondents (10–14, 14–18, >18 years) and one for the rural respondents (10–16 years). Furthermore, all were gender divided for a more accurate account. Next, for each of these intervals, temperature ranges were allocated based on the spot measurements conducted at the time of the survey. The idea behind this approach was to see how thermal comfort is perceived not only depending on the ambient temperature but also based on age. An important remark comes from the number of respondents that took part in the survey. Although the total is 790 (478 male and 312 female), the actual number for which temperature measurements are also available is 635. This discrepancy originates from the way the surveys were distributed and collected. The 155 difference comes from School 4 where, although professors cooperated in handing out surveys during an entire day, measurements were made only during the morning for a limited number of classrooms. With these clarifications out of the way, the attention now shifts to the subject of thermal perception which is evaluated on a scale from very cold (−2) to very warm (+2) with neutral (0) in the middle. All values are expressed in percentages and if added together (within the same temperature range, age interval and gender) make up 100%. This course of action was chosen to avoid any confusion when comparing male and female answers as there was always one gender better represented than the other. 

A few remarks to be made on the data contained in Table 5 would include the following: the best represented age interval is the urban one from 14–18 years (343 respondents), followed by the 10–14 years range (134 respondents), with the last one being the over 18+ years age category (64 respondents). Somewhere in the middle lies the rural 10–16 years interval with 112 respondents. This shows that although there is a clear dominant category, the others are not far apart, ensuring a broad and complete thermal comfort coverage. 

Returning to Table 5, this contains a summary of all the results obtained from the investigation. We will first focus our attention on the answers coming from the urban environment. The first conclusion to be draw is that for all age groups there are very few answers associated with the two extremes (either very cold or very warm) regardless of gender. The only anomaly is registered for male respondents over 18 years of age at an ambient temperature of 23–24 °C. Of them, 44% state that the classroom is very cold despite what common sense would lead us to believe. We can conclude that these are either outliers or insufficiently dressed, and all in all their answers will be disregarded. The vast majority consider the ambient temperature to be satisfactory, regardless of age or gender. For some intervals, female respondents express their contentment at a higher temperature although no trend can be derived. On the other hand, it can be observed that male respondents do prefer somewhat lower temperatures from 20–22 °C. At the same time, this conclusion should be taken with a grain of salt since the sample might be too small to generalize. Additionally, for the 24–24.5 °C span most respondents (44.4% male, 55.6% female) declared that it was warm.

As for the rural respondents, we have a similar trend emerging. This time, the only (expected) difference comes for the 16–18 °C interval. The majority replied that it was either very cold or cold, and fewer stated that it was pleasant. For the rest of the temperature ranges, most answers correspond to the neutral sensation. Overall, it can be stated that at the time of the survey and for the temperatures recorded most respondents were pleased with the classroom temperature with an acceptance level of 82% for the temperature range 18–23 °C (see Figure 5).

The analysis of the data pointed out that 96% of male respondents from rural areas compared to 80% from urban areas are satisfied with a temperature of 20–21 °C while 80% of female respondents from both regions are comfortable within this temperature range (see Table 6). 

It is safe to state at this level that boys from urban environments are more sensitive to the thermal environment than the ones from rural environments. 

Question 2: “In general, during winter, how is the classroom?”

This question presented in the survey attempts to quantify the thermal perception of the respondents during the cold season. Nonetheless, since the surveys were handed out in the middle of autumn, the children had to project themselves into past winters to fill out their answers. This would theoretically induce some errors in the data collected since they are based on a recollection of past experiences which could have faded over time. Regardless of this inconvenience, the survey still manages to paint a qualitative picture of how winter impacts thermal comfort in classrooms. Moreover, our previous experience with measurements of comfort in Romanian schools during the winter season demonstrated that air temperature is found to be in the range 21–24 °C, providing almost full thermal satisfaction of the occupants. Regardless of gender, most of the urban respondents (47.2%) declared that they had a neutral thermal perception during winter, while those in countryside felt that it was warm (54.4%). The same reaction to ambient temperatures continued for the “very cold” response, with more urban respondents complaining about this than their countryside peers (see Figure 6). There could be a twofold explanation for this phenomenon. First, countryside schools could be warmer during winter due to their smaller size and the propensity to burn more wood/natural gas. Generally, in Romanian countryside schools, designated persons fire up the heating wood boiler at 6:00, two hours before the courses start. At 10:00–11:00, the burner is stopped but the thermal inertia of the buildings maintains a comfortable temperature until 13:00–14:00 when the school is closed. The high amount of wood/coal burned since 6:00 increases the indoor air temperature quickly so that from 9:00–11:00 the indoor temperatures could reach 24–25 °C. Moreover, the interior heat gains also contribute to this increase (pupils and lighting representing around 2–3 kW/classroom). A second explanation could come from countryside children being better accustomed to the cold than their colleagues from the cities. However, since the scope of the study did not cover this aspect, it remains only a supposition for now. Lastly, a third explanation would be that countryside children use warmer clothes since they expect that the ambient temperature will be low—especially during the first hours of the morning. The distances between their homes and the school are longer and the time spent on buses/walking is higher than for the students studying in the cities where a larger number schools are in their vicinity.

Question 3: “In general, during summer, how is the classroom?”

The answers given to this third and final question will help assess thermal comfort for the summer period. Although, like for the winter replies, the outcomes are not 100% accurate, they provide clear-cut information on the classroom ambient feel. Since for the warm season it is more difficult to adapt clothing (there is a clothing insulation threshold below which one cannot go), it can be inferred that the general answers given by respondents are more accurate (less likely to be influenced by clothing habits). Figure 7 presents the thermal comfort satisfaction during summer season.

All in all, the results of the subjective investigation point out that the majority of those studying in the city declared that classrooms during summer are very warm (45.9%). As for the differences between genders, this was minimal (2.2%) with boys being affected more by the increased temperatures. A neutral perception was indicated by only 20% of all urban survey replies, whilst a warm perception was highlighted by 31.7%. In addition, based on our team’s expertise on measurement data and in situ investigations it can be stated that there is no air-conditioning in most schools in Romania and for the majority the indoor vertical shading blinds reduce the overheating of the space little. It must also be mentioned that the majority of school buildings built during the communist period are facing east to gain more daylight in the room during morning when students are present. If 50–60 years ago this was a good illumination strategy, nowadays with the increase in temperatures and higher solar irradiation levels the situation is no longer the same and has a clear impact on the overheating of the classrooms. The management and the parents of the children tried to adapt to these conditions and in some classrooms AC units were installed, costs being supported by the parents. As for the replies gathered from the countryside, these offer a somewhat different story. On average, only 23.2% declared that classrooms were very warm during summer, almost half compared to the urban occupants replies. Of respondents, 46.4% thought it was just warm while 19.5% enjoyed a neutral perception during the warm season. It is fair to conclude that based on these findings, children studying in rural schools have fewer complaints during the warm season. The results are to be expected since these schools are in a semi-mountainous region with lower annual average temperatures than the schools situated in a more open field area such as Bucharest. Another reason may be that rural schools, except the old one, had their walls well insulated. All in all, the rural schools performed better than the urban ones during summer.

Jointly taken, all the countryside schools have a better thermal comfort than the urban ones. The explanation comes from having two schools in the countryside which are either renovated or new (including thermal insulation and a newer heating system) whilst the urban schools investigated are not thermally refurbished at the time of the survey. By focusing our attention on the countryside exclusively, the ranking of the schools is the following: School 2 (renovated) is the best performing one, followed by School 6 (new) and ending with School 7 (old). With the previous ranking in mind for this particular case, the conclusion can be drawn that (at least from a perceived thermal comfort perspective) the renovated school is better than the newly constructed one. There are some explanations for this. If the same materials were used to insulate the buildings (10 cm expanded polystyrene for the walls and 20 cm mineral wool for the attic), the main materials of the buildings are not the same. During the communist period, the primary materials used were bricks while nowadays autoclaved cellular concrete is more common. The latter may be cheaper and easier/quicker to build, and the thermal inertia of the buildings is different, where the “winners” are older buildings. Moving on to the answers given for the summer season, across the board it can be observed that all the respondents consider the ambient environment to be warm. The urban schools are consistently warmer than their countryside counterparts. The explanation is quite straightforward since schools that are not mechanically ventilated or air conditioned in a large urban agglomeration will always be warmer (due to the heat island effect of the city) than those located in a green, open, semi-mountainous region. The only noteworthy remark to be made comes from the comparison between the old and new rural schools. The new rural school is warmer than the old one and although located next to each other, a possible explanation for this observation (apart from the subjectivity of those surveyed) may come from the layout of the schools and from the building thermal inertia explained previously. In all cases, old Romanian rural buildings have only a ground floor level and are partially shaded by trees, whilst new schools have an additional first floor and no external shading. In conclusion, the two school categories performed differently when assessed based on the thermal comfort index, depending on the season. During the survey campaign, the rural schools were colder than the urban ones. For winter, the roles changed, only to shift back again for summer (when the warmer buildings were those located in the city). 

### 3.2. Indoor Air Quality

Alongside thermal comfort, air purity plays an equally important role in determining an optimal indoor environment. That is why questions referring to solid particle pollution and overall air quality perception in classrooms were included in the survey. Additionally, a follow-up inquiry on the need to open the windows was added to better correlate the previous two questions. At the same time, spot CO_2_ and particulate level measurements were recorded to better substantiate the answers given and see whether these are consistent with the experimental data. Since the answers that were collected all had different scales, it was decided to create a uniform one ranging from −1 to +1. 

#### 3.2.1. Air Quality in Classrooms

The “air in classroom question” had its answer range reduced to −1 to +1 by adding together the last two options (clean and very clean), to be represented under the value +1. This ensured that when comparing or correlating the answers given to the other questions, the same measurement range could be used. As such, Table 7 highlights the answers to this first initial question and places them under the following headings: urban, rural and overall survey respondents.

Urban, rural and overall survey responses are shown. The number of respondents is included as well as the percent value that this represents from the total number of surveys gathered. All in all, the following conclusion emerges. Rural classroom air quality is by far better than its urban equivalent. Of all rural surveys gathered, 71% declared that the air was “clean”, and only 29% stated that it was “slightly clean”. On the other hand, only 41% of the urban survey respondents indicated that the air was “clean”, 58% that it was “slightly clean” and 8% concluded that it was not clean at all. Furthermore, taken as a whole (both urban and rural settings), there is no definite answer (48% said the air was “slightly clean” and 45% that it was “clean”). The children in the countryside are more satisfied with the air that they breathe inside the classrooms compared to their colleagues from urban schools. The best scores when it comes to air quality are recorded in the new rural school (School 6) and the renovated rural school (School 2). This would imply that the work carried out by the construction team was good or that there are other factors that influenced these good results.

#### 3.2.2. Particle Pollution Present in Classrooms

The second question included in the indoor air quality section of the survey pertained to the perceived dust level. There were three possible answers to choose from: “considerable” (−1), “little” (0) and “not at all” (−1). In addition to this survey question, spot measurements were conducted with the help of the Dylos DC 1100 Pro which records air particles as small as 0.5 microns [41].

We clearly see that on average there is more dust present in the city than in the countryside (see Table 8). Of all urban replies, 57% indicated that there was at least some dust present in the classroom air, 37% stated that there was no dust at all, while 6% complained there was considerable dust. On the other hand, the rural survey responses showed (at least from a perceived level) that there was less dust present in the air. Fifty-three percent could not feel any dust present, 44% indicated there was some and only 3% complained that there was too much dust. These findings will be later correlated with the experimental measurements. The best performing schools (based on this subjective assessment) are School 2 (renovated rural school), School 5 and School 6 (the new rural school). Additionally, the good indoor air quality perception could come from other factors that were not recorded (lower unpleasant odor level, VOC level, etc.). Another logical explanation is that the floors of the classrooms of the old rural school are made from old wood. As such, despite ranking among the lowest on the dust present scale, School 6 could still be accepted to have a good general air quality. It was mentioned earlier that in conjunction with this question, particulate measurements were taken. Using an experimental instrument to assess both fine particles (PM2.5) and coarse particles (PM10) was possible. In Figure 8, the particulate matter index is plotted next to the perceived IAQ index and particles present to better understand whether subjective perception can accurately be used to classify different schools. When focusing on the dust and IAQ index, it can be observed that for six of the eight schools analyzed there is a similar pattern. The perceived IAQ index is always greater than the dust index. This would indicate that children generally consider ambient air to be cleaner than it is although when specifically asked about a particular issue (such as the question referring to the dust level), they tend to be more critical. The remaining two schools show that children consider dust to be less of an issue than overall IAQ. This means that there are other factors (maybe unpleasant odors) that led the respondents to negatively assess IAQ. 

Once the particulate measurements are included in the discussion, the image sketched by the previous two indices becomes clearer. For the schools for which both indices are large (corresponding to a better IAQ), the measurements indicate the lowest particle count (14 particles/cm^3^ in School 2 and 13 particles/cm^3^ in School 6). Coincidence or not, these are also the renovated and new rural schools. For the rest of the schools which have experimental data, we observe the same positive correlation between particulate level and IAQ perception of the respondents. The higher particles’ concentration in the surrounding air, the worse the perception declared. The values presented in Figure 8 for the particle count represent the average total count particles, but on average the PM2.5 values (18.8 particles/cm^3^) were 10.8 times higher than the coarse ones, that is, particles having a mean diameter between 2.5 and 10 microns (1.74 particles/cm^3^).

#### 3.2.3. Need to Open Windows

The last question in the indoor air quality part of the survey was focused on understanding how often the survey participants felt the need to open the classroom windows. Starting from these answers, another index was computed as a weighted average and the results placed on the same −1 to +1 scale (where −1 ≡ rarely; 0 ≡ not often; 1 ≡ often). These indices were directly correlated with the CO_2_ concentration measured at the same time of the surveys distribution. The CO_2_ concentration was recorded for each school and for multiple classrooms and then averaged out before being divided between urban and rural settings. It was evidenced that for all the schools investigated, the average CO_2_ concentration was 1954.6 ppm. Urban schools had on average a lower CO_2_ recording (1883.7 ppm which is 3.6% lower than the global average) than rural schools (which had an average recording of 2285.9 ppm, 16.9% larger than the global average). 

In order to fully understand Table 9 some clarifications are in order. The column “no. of students” contains the respondent’s number by classroom at the time the CO_2_ measurements were made (not every classroom that had children who filled out the surveys also benefitted from CO_2_ recordings). The last column includes the CO_2_ concentration expressed in ppm/m3. Each school had one or more classrooms for which the measurements were conducted, and each classroom was of a certain volume. Initially, the concentration was calculated per classroom and then averaged out for each school. It thus gave us an indication of the CO_2_ mean level for each individual educational building. Although the recommended norms [42] for CO_2_ propose maximum values of 1000 ppm, all of the schools investigated had recorded values several times as high. Despite this, the best performing schools are School 7 (old rural school) with 1404 ppm and School 8 (Technical University of Civil Engineering Bucharest) with 1230 ppm. The explanation behind these marginally better results comes from the level of exterior infiltrations. Since both buildings have old unsealed wooden windows, fresh air can easily penetrate the classrooms. The worst performing of all eight buildings investigated was found to be School 2 (renovated rural school) with an average 2703 ppm CO_2_ level. One explanation could be the increased airtightness of the exterior envelope following the thermal rehabilitation process. At the same time, when divided by the volume of the classrooms the ranking changes only for the high end of the interval. School 5 becomes the worst performing (28 ppm/m^3^) while Schools 1, 2, 3, 4 and 6 have on average 14 ppm/m^3^.

Following these observations, Figure 9 correlates the need to open the classroom windows with the CO_2_ concentration levels. When computing, a correlation coefficient of 0.594 is found, indicating a high likelihood of positive correlation. This means that when the CO_2_ concentration is high, the children instinctively feel the need to open the windows to let fresh air in and thus dilute the gas concentration. The best examples are School 5, where both index and the concentration are high, and School 7, where both index and concentration are low. Of course, these values are not perfectly positively correlated but still the values obtained are to some extent conclusive.

### 3.3. Acoustic Comfort

When discussing indoor environment quality, it is impossible to overlook the part related to acoustic comfort and this survey is no exception. It has a dedicated section comprising two questions, namely, how much noise is coming from the exterior and if the students are disturbed and whether the professor is clearly heard during class hours. Three answers were possible (“considerable”, “little” and “not at all”) in the survey.

As shown in Table 10, the overall perception regardless of urban or rural environment is that the noise coming from the exterior is limited and does not interfere with the class hours. Taken together, the answers “little” and “not at all” make up 85% of all replies for the urban setting and 78% for the rural one. The marginally lower score for the rural schools comes from School 2 being situated next to a county road where heavy traffic is present daily. That is why more rural respondents (22% compared to 15%) complained about considerable noise coming from the exterior. An increased noise level is found for School 2 which was expected since the building is located next to a high-traffic county road. Furthermore, School 1 and School 5 are the best performing buildings given the acoustic comfort criterion. No other surprises are to be found. Using a class 1 sound meter, measurements were realized in order to better understand the responses; thus, it was found out that there is a clear correlation. In Figure 10 is traced the global A-weighted sound pressure level for School 2 confirming that 32% of the respondents in this school found the indoor noise level considerable. It can be observed that at some moments the outdoor sound pressure levels reach even 82 dB(A) and an average of 52 dB(A). Fortunately, the windows of the schools were replaced during the refurbishment and thus reduce the sound waves by more than 16 dB(A) from the average values. The ground floor is more exposed to the outdoor noise and the maximum permissible noise level of 40 dB(A) is surpassed by 23% in the total measurement period while for the 1st floor by 17.7%. The average values are 40.5 dB(A) for the ground floor and 36.3 dB(A) for the 1st floor. 

The next research question to be addressed was whether the level of outside noise had any influence on how well the respondents could hear the professor in class. It seems that for the rural schools the children have no problem hearing what the professor has to say (79.5% declared that they could understand very well what the teacher was saying) despite the exterior noise. One explanation may be that the professors had very strong voices and it was impossible not to hear them. For the urban setting, 43.8% of all respondents were very pleased with how well they could pick up what the professor was saying and only 5.2% had trouble doing so. All in all, despite some exterior noise issues, the children can follow class hours without any problem. The reason for the poor performing schools could be related to the larger classrooms (one of the schools is a university while the other is a technical high school with many reverberant surfaces such as marble floors or large window area) or simply because these students tend to pay less attention to the professor. 

### 3.4. Visual Comfort

The last section of the survey dealt with the visual comfort and contained questions referring to the perceived light level on the workbench and whether the classroom is overall well-lit or not. The answers were placed on the usual −1 to +1 scale (where −1 ≡ insufficient; 0 ≡ suboptimal; +1 ≡ enough bordering excessive) and then correlated with spot illuminance measurements.

In Table 11, the answers to the perceived light level question are summarized. The overwhelming viewpoint is that the illuminance is well and truly sufficient for class activities. To give the reader an example, in the urban schools 93% of those who took part in the survey were satisfied, whereas in the rural schools 95% considered the lux level to be adequate. The differences between genders are minute, with the only engaging comment coming from the fact that rural females are more satisfied by the light level than their urban counterparts and even than urban male students. 

### 3.5. Comparative Analysis

After discussing all the different sections of the survey, this last part of the analysis will focus on providing a ranking mechanism for the schools investigated and then analyzing the validity of the results. One important part in this section is the last question from the survey: weighting the thermal, acoustic and visual comfort and air quality. Our results are based on the responses of 790 occupants that ranked these four aspects. Table 12 summarizes the weighting scheme of other authors in comparison with this paper’s survey results.

Table 12 presents the weighting scheme that is affected by the type of building and by the number/age/sex/occupation of the occupants surveyed. While there may be some discrepancies, one thing is sure if averaging the weights—thermal comfort is among the most important aspects for the occupants while the second is IAQ, followed in the third position by acoustic comfort. In 4th position, just few percent behind the previous three, we found illumination as a main important characteristic of the indoor environment. As multiple factors can influence the IEQ, a general weighting scheme valid for all types of buildings is almost impossible. 

## 4. Conclusions

Maintaining an adequate indoor environment in schools has slowly been recognized as a contributing factor to the learning performance of pupils or their well-being. However, in many cases the IEQ is not achieved as this was proved by measurements or surveys in many papers. When it comes to rural or urban schools, the occupants’ perceptions of the IEQ may be different for many reasons: social class, habitudes, outdoor environment, type of building, etc. Meanwhile, in Romania there is a national program for the rehabilitation of public buildings and schools are among the priorities. Despite the efforts to reduce energy consumption and enhance the indoor environment, there are still many buildings that have not been renovated. Based on the survey realized in multiple Romanian schools with the help of 790 occupant responses, multiple conclusions were obtained. When it comes to thermal comfort, 47% of the urban respondents considered the indoor air temperature as neutral, however, it is safe to state at this level that male respondents from urban environments are more sensitive to the thermal environment than the ones from rural environments. In rural schools, due to poor control of the heating system the indoor temperature quickly rises to more 25 °C, thus 54% of respondents consider it to be a warm environment. For the summer period, the thermal discomfort is much higher for urban respondents explained by either a larger number of occupants per m^2^, urban heat island or higher thermal insulation of the buildings. The air quality seems to be a general problem in all schools but on average there is a higher particle concentration in the city than in the countryside where 57% of all urban replies indicated that there were at least some solid particles present in the classroom air. The results of this study strengthen the evidence that improved ventilation is mandatory in existing schools and that without intervention the existing ventilation rates will remain below the minimum recommended norms. Moreover, in the old rural school the children’s complaints revolved around the presence of dust/particulates and air drafts. Noise is found to be a problem especially for the rural school situated next to a national road. The survey on illuminance levels showed that there is plenty of light for most respondents—in the urban schools, 93% were satisfied, whereas for the rural respondents, 95% considered the lighting level to be adequate. In this paper, a newly developed weighting scheme based on the 790 surveys to classify the indoor environmental quality based on four aspects (light, sound, thermal comfort, air quality) is also presented. In the top position is the indoor air quality, followed by thermal, visual and finally acoustic comfort. The study was limited to eight schools and to only one Romanian regional rural area. The outdoor climatic data and air quality parameters clearly impacted the respondents and further analysis to cover different scenarios (e.g., smaller versus larger cities with different pollution levels in terms of noise, air quality, temperatures) is recommended. The weighting scheme for IEQ proposed in this paper presents new information to the scientific community but additional surveys are strongly recommended in other countries to better validate the choice of target respondents. 

## Figures and Tables

**Figure 1 ijerph-19-10219-f001:**
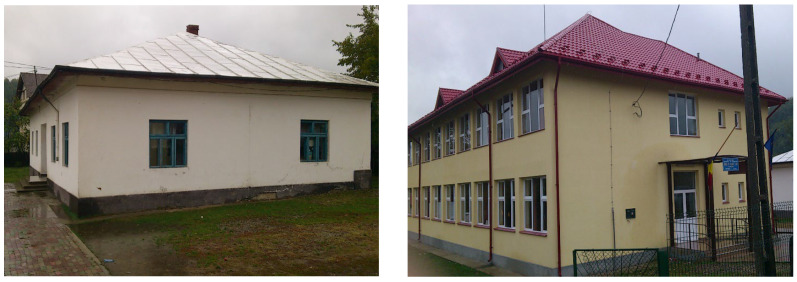
Exterior facades of typical rural/urban schools in Romania.

**Figure 2 ijerph-19-10219-f002:**
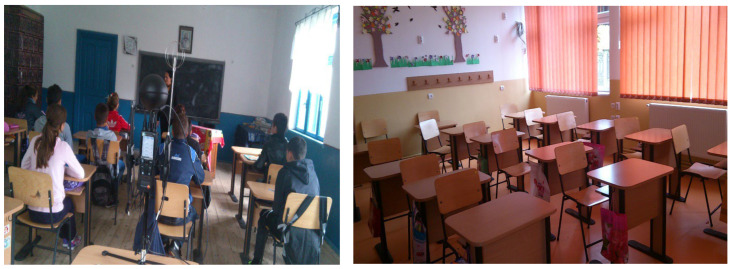
Interior of a typical classroom in old rural region vs. new/renovated school.

**Figure 3 ijerph-19-10219-f003:**
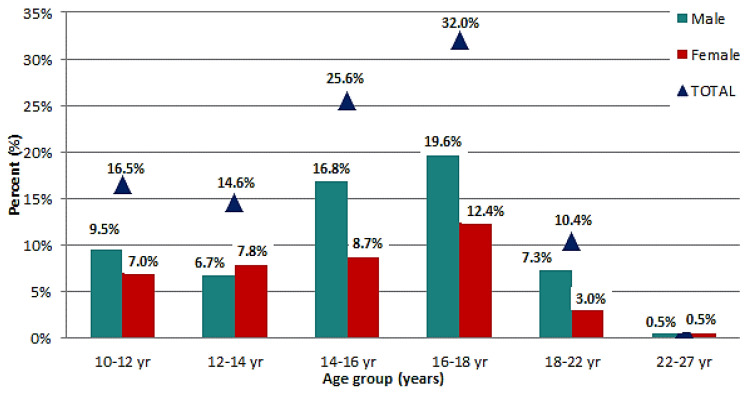
Age distribution of survey respondents.

**Figure 4 ijerph-19-10219-f004:**
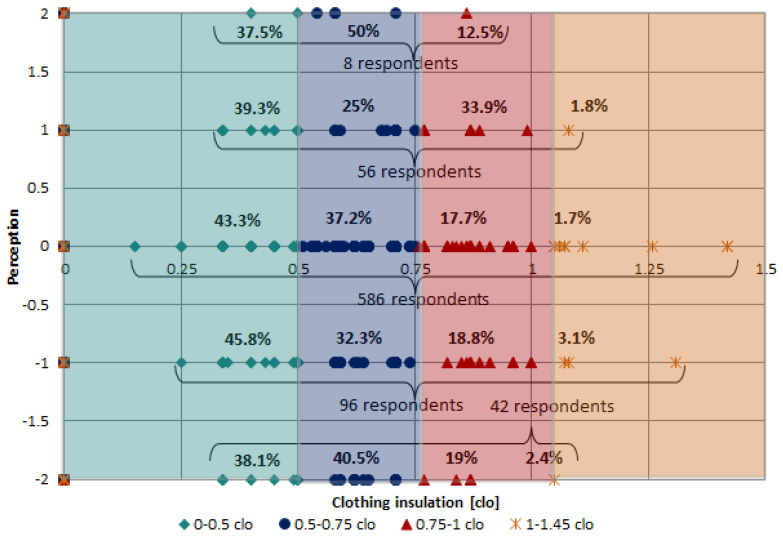
Clothing insulation vs. thermal comfort perception.

**Figure 5 ijerph-19-10219-f005:**
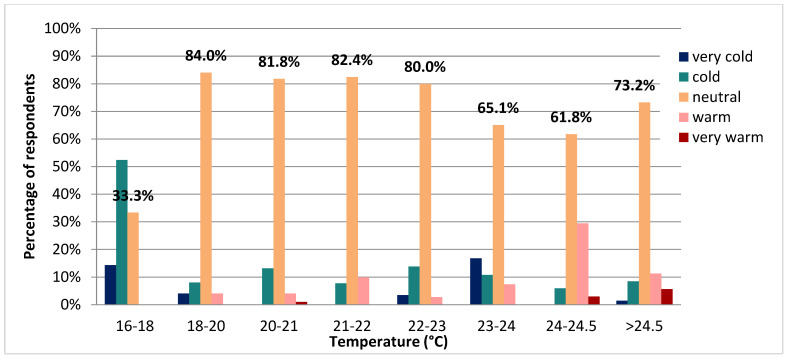
Thermal perception overall of survey correlated with the measured air temperatures.

**Figure 6 ijerph-19-10219-f006:**
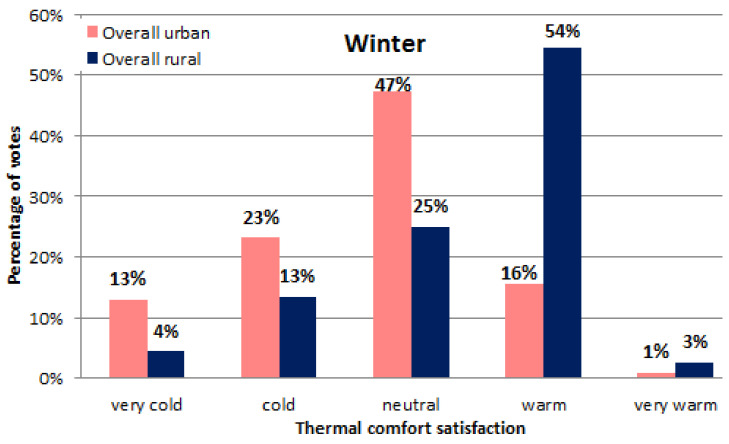
Winter thermal satisfaction (urban vs. rural schools).

**Figure 7 ijerph-19-10219-f007:**
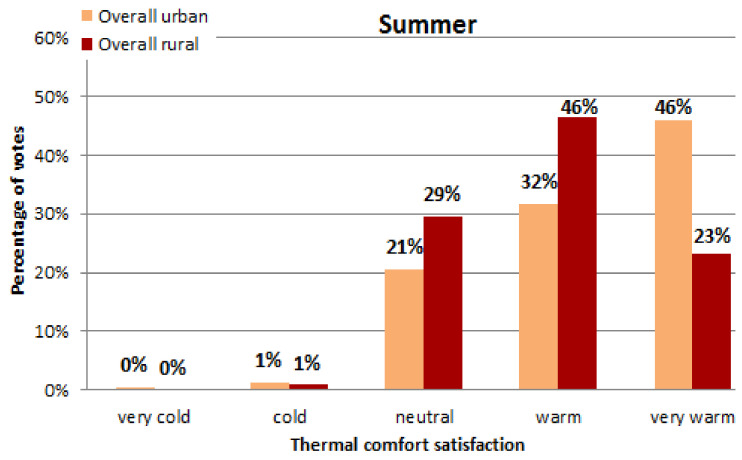
Summer thermal satisfaction (urban vs. rural schools).

**Figure 8 ijerph-19-10219-f008:**
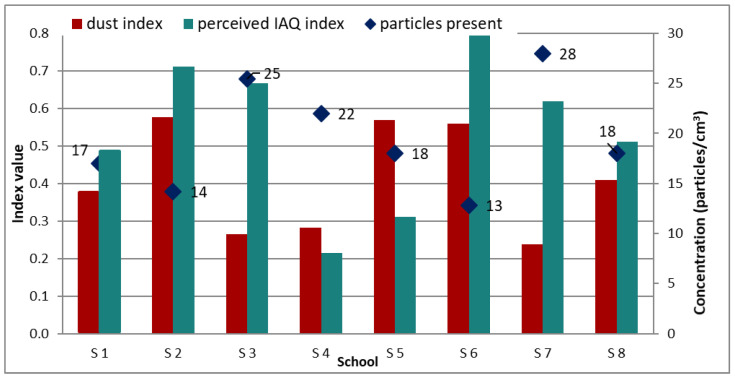
Dust index correlated with perceived IAQ index and particles present.

**Figure 9 ijerph-19-10219-f009:**
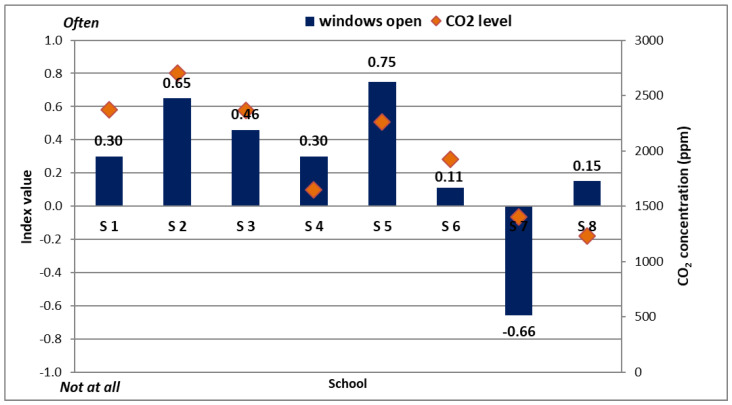
Need to open the windows vs. CO_2_ concentration.

**Figure 10 ijerph-19-10219-f010:**
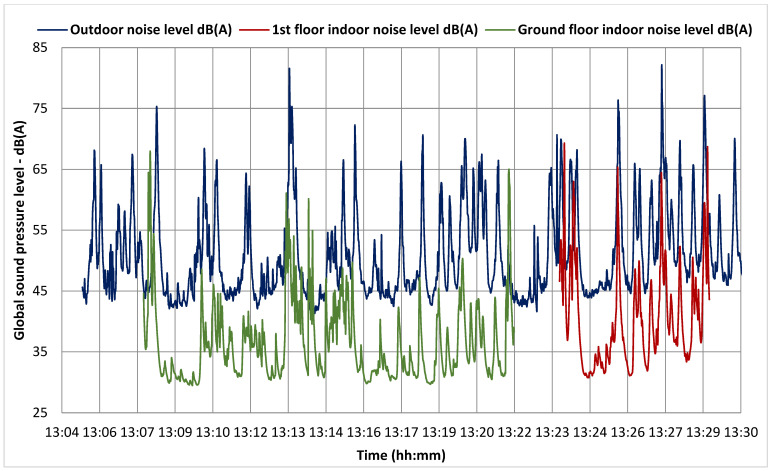
Global sound pressure level measurements for School 2.

**Table 1 ijerph-19-10219-t001:** Equipment specifications.

Probe	Variables	Meas. Range	Resolution
Testo 480	Humidity and air temperature	−100 to +400 °C	0.1 °C
	Air velocity	0 to +20 m/s	0.01 m/s
	CO_2_ level	0 to 100 %rH	0.1 %rH
	Radiant temperature	0 to 10,000 ppm CO_2_	1 ppm CO_2_
	Illuminance level	0 to 100,000 Lux	1 lux
Bruel and Kjaer 2250	Sound pressure level	1.1 to 140 dB	Class 1 precision 0.1 dB
TSI 7545	CO_2_ level indoor CO level Wet bulb temperature Dew point temperature	0 to 5000 ppm 0 to 500 ppm 0 to 60 °C	1 ppm 0.1 ppm ±0.6 °C
Dylos 1100 Pro	Particulate	0–200,000 #/L	−325% to 78%
CO_2_ m	CO_2_ level indoor/outdoor	0–10,000 ppm	1 ppm
	Indoor air temperature/humidity	0 to 100 %rH	0.3 %rH
	Outdoor air temperature/humidity	−40 to +60 °C	0.4 °C

**Table 2 ijerph-19-10219-t002:** Educational buildings surveyed.

School Name	Tag	Location	Year of Construction	Number of Respondents	Percent of Total
“*George Cosbuc*” National College	School 1	Urban	1972	90	11.4%
“Mateesti” Primary School	School 2	Rural	1960	66	8.4%
“*Liviu Rebreanu*” Primary School	School 3	Urban	1965	72	9.1%
“*Anghel Saligny*” Technical College	School 4	Urban	1966	419	53.0%
“*Special school*” Primary School	School 5	Urban	1960	45	5.7%
“Turcesti” Primary School 1	School 6	Rural	2005	25	3.2%
“Turcesti” Primary School 2	School 7	Rural	1940	23	2.9%
UTCB Technical University	School 8	Urban	1948	50	6.3%

**Table 3 ijerph-19-10219-t003:** Survey population.

Age (Years)	Urban			Rural		
	Total	Male	Female	Total	Male	Female
10–12	68	39	29	62	36	26
12–14	66	25	41	49	28	21
14–16	201	132	69	1	1	-
16–18	254	155	98	-	-	-
18–22	82	58	24	-	-	-
22–27	8	4	4	-	-	-

**Table 4 ijerph-19-10219-t004:** Clothing insulation proposed in the survey.

Clothing Type	Insulation Value (clo)	Clothing Type	Insulation Value (clo)
Long trousers	0.25	Down jacket	0.55
Shorts	0.06	Sweater	0.37
Shirt	0.20	Coat	0.60
Hoodie	0.37	Light jacket	0.25
Skirt	0.18	Leggings	0.10
Light blouse	0.15	Jacket	0.35
Normal blouse	0.25	Jeans	0.28
Short sleeve shirt	0.09	Vest	0.10
Gym wear	0.49	Sleeveless shirt	0.06

**Table 5 ijerph-19-10219-t005:** Survey respondents’ classification based on age, gender and temperature interval.

Age	Temperature Range (°C)	URBAN			Age Interval	Temperature Range (°C)	RURAL		
		Male	Female	Total			Male	Female	Total
**10–14 years**	20–21	14	10	24	**10–16 years**	16–18	10	11	21
	21–22	23	19	42		18–20	14	11	25
	22–23	8	14	22		20–21	29	20	49
	23–24	10	9	19		21–22	12	5	17
	24–24.5	4	5	9	**#N/A**				
	>24.5	4	14	18					
**14–18 years**	20–21	11	15	26					
	21–22	10	10	20					
	22–23	51	29	80					
	23–24	87	34	121					
	24–24.5	7	17	25					
	>24.5	26	27	53					
**>18 years**	20–21	11	1	12					
	21–22	28	15	43					
	22–23	8	1	9					

**Table 6 ijerph-19-10219-t006:** Comparison of thermal perception of the survey respondents from urban and rural environments.

Age	Temp. (°C)	Thermal Perception (Urban Respondents)									
		Very Cold		Cold		Neutral		Warm		Very Warm	
		M	F	M	F	M	F	M	F	M	F
**10–14 years**	20–21	0.0%	0.0%	8.3%	4.2%	45.8%	33.3%	4.2%	4.2%	0.0%	0.0%
	21–22	0.0%	0.0%	4.8%	2.4%	50.0%	35.7%	0.0%	7.1%	0.0%	0.0%
	22–23	0.0%	0.0%	9.1%	4.5%	22.8%	59.1%	4.5%	0.0%	0.0%	0.0%
	23–24	0.0%	0.0%	0.0%	0.0%	47.4%	31.6%	5.3%	15.7%	0.0%	0.0%
	24–24.5	0.0%	0.0%	0.0%	0.0%	0.0%	0.0%	44.4%	55.6%	0.0%	0.0%
	>24.5	0.0%	0.0%	5.6%	0.0%	5.6%	61.1%	11.0%	5.6%	0.0%	11.1%
**14–18 years**	20–21	0.0%	0.0%	3.8%	23.1%	34.7%	34.6%	3.8%	0.0%	0.0%	0.0%
	21–22	0.0%	0.0%	0.0%	5.0%	40.0%	45.0%	10.0%	0.0%	0.0%	0.0%
	22–23	2.5%	1.3%	10.0%	6.3%	48.8%	27.5%	2.5%	1.3%	0.0%	0.0%
	23–24	9.9%	7.4%	9.9%	3.3%	49.6%	14.9%	2.5%	2.5%	0.0%	0.0%
	24–24.5	0.0%	0.0%	8.0%	0.0%	24.0%	60.0%	0.0%	4.0%	0.0%	4.0%
	>24.5	0.0%	1.9%	5.7%	3.8%	38.6%	35.8%	1.4%	7.2%	1.9%	3.7%
**>18 years**	21–22	0.0%	0.0%	8.3%	0.0%	75.0%	8.4%	8.3%	0.0%	0.0%	0.0%
	22–23	2.3%	2.3%	7.0%	2.3%	55.8%	30.3%	0.0%	0.0%	0.0%	0.0%
	23–24	44.4%	0.0%	0.0%	0.0%	33.3%	11.2%	11.1%	0.0%	0.0%	0.0%
		**Thermal Perception (Rural Respondents)**									
**10–16 years**	16–18	4.8%	9.5%	28.6%	23.8%	14.3%	19.0%	0.0%	0.0%	0.0%	0.0%
	18–20	4.0%	0.0%	8.0%	0.0%	44.0%	40.0%	0.0%	4.0%	0.0	0.0%
	20–21	0.0%	0.0%	0.0%	6.1%	57.1%	32.8%	0.0%	2.0%	2.0%	0.0%
	21–22	0.0%	0.0%	5.9%	5.9%	47.1%	23.5%	17.6%	0.0%	0.0%	0.0%

**Table 7 ijerph-19-10219-t007:** General air quality perception.

Answer	Urban			Rural			Overall		
	Mixed	Male	Female	Mixed	Male	Female	Mixed	Male	Female
**Clean (1)**	40.6%	37.8%	45.1%	71.4%	67.7%	76.6%	45.0%	41.8%	49.8%
**Slightly clean (0)**	51.4%	53.8%	47.7%	28.6%	32.3%	23.4%	48.2%	50.8%	44.1%
**Not clean at all (−1)**	8.0%	8.5%	7.2%	0.0%	0.0%	0.0%	6.8%	7.3%	6.1%

**Table 8 ijerph-19-10219-t008:** Perception of the dust level present in classrooms.

Answer	Urban			Rural			Overall		
	Mixed	Male	Female	Mixed	Male	Female	Mixed	Male	Female
**Considerable**	5.5%	5.1%	6.1%	2.7%	3.1%	2.1%	5.1%	4.8%	5.5%
**Little**	57.4%	61.3%	51.3%	43.8%	44.6%	42.6%	55.5%	59.0%	50.0%
**Not at all**	37.1%	33.7%	42.6%	53.6%	52.3%	55.3%	39.5%	36.2%	44.5%

**Table 9 ijerph-19-10219-t009:** CO_2_ concentration per school/student/m^3^.

Location	No. of Students	[ppm]	STDEV	[ppm/Student]
School 1	88	2369.9	765.7	189
School 2	66	2703.3	945.4	152
School 3	71	2367.4	886.8	18
School 4	25	1644.7	604.4	93
School 5	9	2259.4	638.0	267
School 6	25	1924.3	59.1	154
School 7	21	1404.1	220.9	152
School 8	35	1230.3	755.4	63

**Table 10 ijerph-19-10219-t010:** Perception of the outdoor noise disturbance.

Answer	Urban			Rural			Overall		
	Mixed	Male	Female	Mixed	Male	Female	Mixed	Male	Female
**Considerable**	14.7%	18.6%	8.7%	22.3%	21.5%	23.4%	15.8%	19.0%	10.9%
**Little**	56.9%	57.4%	56.2%	54.5%	58.5%	48.9%	56.6%	57.5%	55.1%
**Not at all**	28.3%	24.0%	35.1%	23.2%	20.0%	27.7%	27.6%	23.4%	34.0%

**Table 11 ijerph-19-10219-t011:** Perception of the indoor light level.

Visual Comfort	Urban			Rural			Overall		
	Mixed	Male	Female	Mixed	Male	Female	Mixed	Male	Female
Enough	93.5%	94.9%	91.3%	95.5%	92.3%	100.0%	93.8%	94.6%	92.6%
Suboptimal	5.8%	4.1%	8.3%	2.7%	4.6%	0.0%	5.3%	4.2%	7.1%
Insufficient	0.7%	1.0%	0.4%	1.8%	3.1%	0.0%	0.9%	1.3%	0.3%

**Table 12 ijerph-19-10219-t012:** Summary of IEQ category weighting schemes.

Study	Type of Building	No. Occupants Surveyed	Acoustics	IAQ	Lighting	Thermal Comfort
[43]	Dwelling	12	0.23	0.34	0.19	0.24
[44]	Office	293	0.24	0.25	0.19	0.31
[45]	Healthcare	-	0.25	0.176	0.23	0.38
[46]	Public buildings	500	0.27	0.14	0.21	0.38
[47]	Office	68	0.18	0.36	0.16	0.30
[27]	Commercial	52,980	0.39	0.20	0.29	0.12
** *Proposed values* **	**School**	**790**	**0.19**	**0.30**	**0.24**	**0.27**

## Data Availability

Not applicable.

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
