# Peer review of "Survey and Measurements of Indoor Environmental Quality in Urban/Rural Schools Located in Romania"

_ijerph, 2022, doi:10.3390/ijerph191610219_

Round 1

Reviewer 1 Report

This paper studies the indoor environmental quality in atmospheric air quality, basically the Romanian schools. The following problems need to be addressed:

1. Line 366: “for the period 30/09/2013-08/10/2013”. If that is the case, I would reject this paper because the data are outdated.

2. The organization of this paper is very strange. Following “2.Methodology” is “3.Conclusion”, not “Results/Discussions”.

3. There are no deep analyses for the results.

4. The authors need to polish the language. In addition, there are many typos and editing problems.

5. There is “during cold season” in the title. However, both summer season and winter season are studied.

6. Which season did the authors study for IAQ?Why is it different from thermal comfort?

7. Line 405: Why are PM 2.5 values higher than PM10?

8. Table 10: I don’t think the values for the last column is very meaningful. The unit is kind of strange as well.

9. Table 5 and 6: for “>18 years”, the temperature ranges are not the same.

Author Response

The authors would like to thank you for the time spent reviewing our paper. Your comments are useful, and we have now an enhanced version of our paper.

 Point 1: Line 366: “for the period 30/09/2013-08/10/2013”. If that is the case, I would reject this paper because the data are outdated.

 Response 1: The informations are recent so we erased the mistaken period of year

Point 2: The organization of this paper is very strange. Following “2.Methodology” is “3.Conclusion”, not “Results/Discussions”.

Response 2: You are right. We have changed it now to be Introduction – Methodology – Discussion (with different sub-chapters) --- Conclusions

Point 3: There are no deep analyses for the results.

Response 3: The surveyed data was based on 790 occupants’ responses (children) and was quite a challenge. In fact, this kind of survey with children is among the largest at international level. The survey was completed by experimental measurements using multiple top measurements equipment’s (Table 1). The analysis is deep in the way we treated the whole survey – even we have presented data on their clothing at the time of survey (Table 4) and we made an interesting analysis on clothing insulations expressed in clo (calculated based on their clothing) versus thermal perception in classroom. There are other interesting points like thermal perception correlated with measured data (Figure 5). Overall, we believe there are many interesting results that may trigger attention for the scientific community.

 Point 4. The authors need to polish the language. In addition, there are many typos and editing problems.

Response 4: Indeed, we have found multiple errors and we have corrected them. The paper was written by Alexandru GHITA with a high level of English at Delft, University. The paper has more than 11 000 words in 25 pages and inevitably there were errors.

Point 5. There is “during cold season” in the title. However, both summer season and winter season are studied.

Response 5: We modified the title to be more exact with the data presented. The title was changed to,, Survey and measurements of indoor environmental quality in urban/rural schools located in Romania,,

Point 6: Which season did the authors study for IAQ?Why is it different from thermal comfort?

Response 6: The survey on indoor air quality was realized during winter time when the time spent inside is much larger than summer. At the same time questions about thermal comfort were also asked. We found important also to ask about summer discomfort – this is a well-known problem that many classrooms have during the month of June and September (July and august are holidays). To resume the answer – the IAQ was measured/rated during winter time when is the most critical while thermal comfort was analyzed in the same time but also we had questions about summer discomfort.

Point 7 Line 405: Why are PM 2.5 values higher than PM10?

Response 7: In general, smaller particles like the 2.5 were found to be higher in all measured schools. Most probably is due to the fact of using chalk as mean of writing on the blackboard.

Point 8 Table 10: I don’t think the values for the last column is very meaningful. The unit is kind of strange as well.

Response 8: Indeed, it is not common to use ppm/m3 so we erased that column. Thank you for the remark.

Point 9. Table 5 and 6: for “>18 years”, the temperature ranges are not the same.

 Response 9: Yes, you are right. It was a small typo error – both recordings where between 22-24 degree Celsius

Reviewer 2 Report

The Abstract of the article is interesting however, it should be improved. It would be important that the authors identify which Research Methodology the authors followed, as well as what is the research question.

I recommend that the authors change the keyword "questionnaire" to "survey", it makes more sense.

[line 91-92] Authors cannot write the standard as "The EN Standard 1264-1 establishes...", but rather as "The Standard EN 1264-1 establishes...".

[line 100-101] Who is Mr 27? "... and cause absence from school as it was found by [27] in elementary schools in Finland." - Clearly there is an error in the referencing system here that the authors should review. These situations should be reviewed and corrected!

[line 136] Why did the authors put a full stop before reference, and another one just after: "... vey fatigue".[35]. "Right-now" surveys ..."??? The authors should review this situation and correct it!

When authors put the reference at the end of the sentence, the full stop should always come after the reference, and never before. Authors should review and correct, for example, the line references "[line 94] … lighting. [26]", and also "[line 110-111] … educational establishments. [3]]" and "[124-125] … CBE Berkly University one. [33]", among many others throughout the article.

It should be clarified by the authors how the sample for the study of this research was selected. This information is very relevant for the reliability of this research.

The authors did not identify what the limitations of this research were and should review and add this information to the article.

This research is interesting, but it would be very good if the authors could recommend future work, so that other authors could follow up on this research with other publications.

Author Response

The authors would like to thank you for the time spent reviewing our paper. Your comments are useful and we have now an enhanced version of our paper.

 Point 1: The Abstract of the article is interesting however, it should be improved. It would be important that the authors identify which Research Methodology the authors followed, as well as what is the research question.

 Response 1: The abstract was modified.

 Point 2: I recommend that the authors change the keyword "questionnaire" to "survey", it makes more sense.

 Response 2: Thank you for the suggestion. We have modified the entire document.

Point 3: [line 91-92] Authors cannot write the standard as "The EN Standard 1264-1 establishes...", but rather as "The Standard EN 1264-1 establishes...".

Response 2: We have modified.

Point 3: [line 100-101] Who is Mr 27? "... and cause absence from school as it was found by [27] in elementary schools in Finland." - Clearly there is an error in the referencing system here that the authors should review. These situations should be reviewed and corrected!

Response 3: Yes, indeed. We have corrected the reference 27 and also other ones for better referencing mentioning the authors.

 Point 4. When authors put the reference at the end of the sentence, the full stop should always come after the reference, and never before. Authors should review and correct, for example, the line references "[line 94] … lighting. [26]", and also "[line 110-111] … educational establishments. [3]]" and "[124-125] … CBE Berkly University one. [33]", among many others throughout the article.

Response 4: We have corrected theses issues.  

Point 5. It should be clarified by the authors how the sample for the study of this research was selected. This information is very relevant for the reliability of this research.

Response 5: We have added 78 words in a paragraph to clarify this part.

Point 6: The authors did not identify what the limitations of this research were and should review and add this information to the article.

Response 6: We have added an entire paragraph in the conclusions

Point 7 This research is interesting, but it would be very good if the authors could recommend future work, so that other authors could follow up on this research with other publications.

Response 7: We have added an entire paragraph in the conclusions

Round 2

Reviewer 1 Report

The authors addressed most of my concerns. There is still one problem.

"Why are PM 2.5 values higher than PM10? Response 7: In general, smaller particles like the 2.5 were found to be higher in all measured schools."

From the concept of PM2.5 and PM10, The value of PM10 is always higher than the value of PM2.5.

Author Response

Indeed, by definition PM10 represents fine and coarse particles, determined by a mean diameter less than 10 microns, so PM2.5 represents part of PM10. The authors should explain that: “The higher particles’ concentration in the surrounding air, the worse is the perception declared. The values presented in Figure 9 for the particles count represent the average total count particles, but on average the PM2.5 values (18.8 particles/cm3) were 10.8 times higher than the coarse ones, particles having the mean diameter between 2.5 and 10 microns (1.74 particles/cm3).” The change was also made in the manuscript.

Reviewer 2 Report

The article has been revised and strongly improved by the authors. Congratulations to the authors for their work!

Author Response

Thank you for the time spent on this paper. Best regards